# COVID-19 Vaccination Uptake and Related Determinants in Detained Subjects in Italy

**DOI:** 10.3390/vaccines10050673

**Published:** 2022-04-23

**Authors:** Gabriella Di Giuseppe, Concetta Paola Pelullo, Raffaele Lanzano, Chiara Lombardi, Giuseppe Nese, Maria Pavia

**Affiliations:** 1Department of Experimental Medicine, University of Campania “Luigi Vanvitelli”, Via Luciano Armanni 5, 80138 Naples, Italy; gabriella.digiuseppe@unicampania.it (G.D.G.); concettapaola.pelullo@uniparthenope.it (C.P.P.); raffaele.lanzano@studenti.unicampania.it (R.L.); chiara.lombardi@studenti.unicampania.it (C.L.); 2Department of Movement Sciences and Wellbeing, University of Naples “Parthenope”, Via Medina 40, 80133 Naples, Italy; 3U.O.C. Tutela Salute in Carcere—ASL Caserta, Via Unità Italiana 28, 81100 Caserta, Italy; giuseppe.nese@aslcaserta.it

**Keywords:** attitudes, COVID-19 vaccination, incarcerated people, Italy, survey, uptake

## Abstract

Background: This study explored the extent of COVID-19 vaccination coverage and investigated drivers and barriers to COVID-19 vaccine uptake among people in prison. Methods: This cross-sectional study was conducted from July to October 2021 among 517 detained people in the Campania region of South Italy. Results: In total, 47.1% of participants expressed a high concern about contracting COVID-19 after vaccination, whereas 60.6% and 53.8% of respondents reported a positive attitude towards usefulness and safety of COVID-19 vaccines, respectively. Adherence to the active offer of COVID-19 vaccination involved 89.7% of detained subjects. COVID-19 vaccination uptake was significantly higher in females, and in those who reported influenza vaccination uptake, had received information about COVID-19 vaccination from media and newspapers, did not express need of additional information about COVID-19 vaccine, believed that COVID-19 vaccine is safe, were involved in working activities in the prison, and had a high school or university degree. Conclusions: These findings showed a high self-reported COVID-19 vaccination coverage in detained subjects, supporting the effectiveness of the strategy aimed at giving priority to COVID-19 vaccinations in prisons. Further efforts are needed to contrast the hesitancy of those who refused vaccination to increase their confidence about usefulness and safety of COVID-19 vaccines.

## 1. Introduction

Since the beginning of the COVID-19 pandemic, prisons have been recognized as high-risk settings, showing higher incidence and mortality rates than the outside community. According to the most updated data, from October 2020 to October 2021, the cumulative COVID-19 cases occurred in European countries prisons were 29,741, with an incidence of 11,941 COVID-19 cases/100,000 people in prison; and 187 cumulative COVID-19 related deaths [1]. In Italy, the Ministry of Justice has set up a weekly report on COVID-19 cases in Italian prisons, showing a relatively low frequency during the first wave, not exceeding 160 weekly positive prisoners and a total 4 deaths, which steadily increased during the second wave with more than 1000 positive subjects per week and 16 deaths. From January to March 2021, the number of infections fell again, with a weekly rate of about 480 cases, and a total of 18 deaths. The latest available data, updated to 12 April 2022, report 1080 COVID-19 positive out of 53,945 incarcerated people, with 1078 asymptomatic and 2 hospitalized subjects [2]. Despite these variable frequencies, the COVID-19 incidence rate among incarcerated subjects has always been higher than that reported in the general population. Specifically, in April 2020, the incidence in Italian prisons was 18.7 per 10,000 people versus 16.8 per 10,000 people in the general population, in December 2020 179.3 versus 110.5, and in February 2021 91.1 versus 68.3 [3].

The surging of COVID-19 cases and outbreaks in prisons all over the world has prompted the World Health Organization (WHO) to develop guidelines to promote preparedness, prevention, and control of COVID-19 in prisons in the first months of the pandemic [4], that have been updated in 2021, as new evidence on the evolving epidemic and the ways to contrast it were becoming available [5].

With the approval of COVID-19 vaccines, countries all over the world have been involved in the design of vaccination strategies, that were initially targeted to subgroups of the population showing higher risk of SARS-CoV-2 exposure and transmission, and of more severe COVID-19 clinical course. In this context, several organizations and agencies, considering that the response to COVID-19 in places of detention is particularly challenging, that people in prison have increased vulnerability to infectious diseases and COVID-19, and a higher burden of comorbidities leading to increased risk of severe outcomes, advocated the importance of including detention facilities as a priority setting for COVID-19 vaccination plans [6,7,8,9,10]. This is notably of relevance in Italy, since in 2020 it was second after Cyprus, as the European country with the most overcrowded prisons [11]. Moreover, data updated on 31 March 2022 reveal that the total number of inmates is 54,609, and the total capacity of penal institutions is 50,853 [12] with a turnover ratio of 50.1 [13].

In Italy the COVID-19 vaccination strategy set by the Government introduced a step-by-step involvement of groups of the population, based on their risk of infection and of severe clinical course, and people in prison and correctional facilities staff were included as a priority group to be vaccinated [14,15,16]. When the study was conducted, the following vaccines were already authorized and used in the COVID-19 vaccination campaign: two mRNA vaccines namely Comirnaty (Pfizer/BioNTech), with an efficacy of 95% against symptomatic SARS-CoV-2 infection, and Spikevax (Moderna), with an efficacy of approximately 94.1% in protecting against COVID-19; one recombinant viral vector vaccine, Vaxzevria (AstraZeneca), with an efficacy against symptomatic SARS-CoV-2 infection of 76%; and one viral vector vaccine, COVID-19 Vaccine Janssen (Janssen Cilag), with an efficacy of 85.4% against critical illness and 93.1% against hospitalization [17].

Since monitoring of vaccine knowledge, attitudes, and behaviors has been placed as a strategic basis for all vaccination programs [18], and the study of the determinants of vaccination choice as one of the guiding tools for national immunization programs and for the development of specific strategies for the improvement of coverage [18], several studies have been conducted to explore the willingness of people in prison to undergo COVID-19 vaccination, as well as determinants of intention to uptake the vaccine [19,20,21]. However, despite the contribution of studies on willingness and related determinants is undoubtedly relevant, it is known that intention to undergo vaccination may not always predict real vaccination uptake [22], and identification of determinants of hesitancy among those who delay or refuse vaccinations is a key component in the design of interventions aimed at the achievement of successful vaccination coverage.

Therefore, the aim of this survey was to explore the extent of COVID-19 vaccination coverage and to investigate drivers and barriers to COVID-19 vaccine uptake among people in prison in Italy.

## 2. Materials and Methods

### 2.1. Study Design and Sampling

This survey was carried out between July and October 2021, within a larger project developed by the University of Campania “Luigi Vanvitelli” and the Joint Operational Unit for Health Protection at Prison Institutions, investigating several health-related issues in the incarcerated population [21,23,24]. Three prisons in the geographic area of Campania region, in the South of Italy, were included in the study.

For the sample recruitment, detained people were first stratified by their detention status, based on low, medium, or high security regimes. Then, a proportional number of subjects were randomly selected from each group and invited to participate to the survey.

According to the existing literature exploring the uptake of COVID-19 vaccination, the hypothesized proportion of adhesion was fixed at 65%; therefore, setting the alpha error at 5% and the confidence interval at 95%, the required sample size was 437 subjects.

### 2.2. Data Collection

An invitation letter to present the study protocol was sent to all directors of prisons to obtain their consent to conduct the survey. The detained subjects completed a self-administered questionnaire in their prison cell. Those who were allocated in special housing units or were unable to give informed consent because of substantial cognitive impairment or to read and comprehend Italian language were excluded from the survey. The selected subjects were previously informed about survey aims, that the participation was voluntary and confidential, and that no personal information was included in the questionnaire.

### 2.3. Survey Questionnaire

Based on an extensive literature review, the outline and the content of the questionnaire were developed from several validated sources, used to investigate COVID-19 vaccination in detained subjects [21,25], in other populations [26,27,28,29], or other health-related issues in detained subjects [24]. Specifically, the questions to investigate the main socio-demographic, anamnestic, and detention characteristics were extrapolated from the studies by Di Giuseppe et al. [21] and Chin et al. [25], those relating to the main reasons for refusing vaccination from the studies by Stern et al. [19] and Di Giuseppe et al. [26], whereas attitudes towards COVID-19 and behaviors related to the main preventive measures and sources of information were based on the study by Della Polla et al. [29].

The questionnaire was composed of five sections. The first was on socio-demographic and detention characteristics, including age, nationality, marital status, number of sons/daughters, education level, employment status before and during detention, first or multiple episodes of detention, and living in individual or shared cells. The second explored anamnestic characteristics, such as presence and type of chronic diseases, previous SARS-CoV2 infection, history of previous vaccination and uptake of COVID-19 vaccine and reasons for vaccination uptake or refusal. The third section was on attitudes, such as perception of COVID-19 vaccine usefulness and safety, and, among vaccinated subjects, concerns about contracting COVID-19 disease. The fourth was on behaviors after COVID-19 vaccination, such as wearing masks and performing hand antisepsis. The last investigated sources and needs of information about COVID-19 vaccination.

The response choices for sociodemographic, detention, or anamnestic characteristics were close ended with “yes” or “no” or multiple choices response format; those on attitudes were on a ten-point Likert-type scale ranging from 1 for not at all concerned/useful/safe to 10 for very concerned/useful/safe. Questions pertaining to behaviors were close ended with “yes” or “no” or multiple choices or on five-point Likert-type scale with “never” “rarely” “sometimes” “often” and “always” response format.

A pilot study was conducted on 50 detained subjects to improve understanding and interpretation of the questions.

### 2.4. Ethical Statement

The study was approved by the Ethics Committee “Campania Nord” of the Local Health Unit of Caserta (protocol code: 400).

### 2.5. Statistical Analysis

All statistical analyses were performed using STATA software, version 15.0 [30]. Bivariate appropriate tests, *t*-test for continuous variables and Chi-square test for dichotomous or categorical variables, have been conducted to assess the univariate associations between the independent characteristics and COVID-19 vaccination uptake. Then, a multivariate stepwise logistic model was performed to identify the role of several potential determinants associated with COVID-19 vaccination uptake among incarcerated people (no = 0; yes = 1). Adjusted odds ratio (ORs) and 95% confidence intervals (CIs) were calculated. Backward stepwise elimination was applied, and the final model included characteristics that provided a significant explanation of the outcome of interest, setting the criterion for entering and being retained by the stepwise procedure at a *p* value of 0.2 and 0.4, respectively. The level of statistical significance was set at *p* ≤ 0.05. The following independent variables were included in the model: age (19–37 = 1; 38–47 = 2; >47 = 3), gender (male = 0; female = 1), nationality (foreigners = 0; Italian = 1), marital status (unmarried/widowed/separated/divorced = 0; married/cohabitant = 1), having sons/daughters (no = 0; yes = 1), education level (primary school = 1; middle school = 2; high school/university degree = 3), first detention (no = 0; yes = 1), working activities in the prison (no = 0; yes = 1), being vaccinated against influenza in the 2020–2021 season (no = 0; yes = 1), at least one chronic disease (no = 0; yes = 1), belief that COVID-19 vaccine is useful (continuous), belief that COVID-19 vaccine is safe (continuous), having received information about COVID-19 vaccine from physicians (no = 0; yes = 1), having received information about COVID-19 vaccine from media and newspapers (no = 0; yes = 1), and need of additional information about COVID-19 vaccination (no = 0; yes = 1). Adjustment for institution (prison) was also performed.

## 3. Results

### 3.1. Socio-Demographic, Detention, and Anamnestic Characteristics of the Study Population

Of the 650 detained people invited to participate in the study, 517 agreed and returned the survey for an overall response rate of 79.5%. In the three selected institutions, between 2020 and 2022, 669 COVID-19 cases were reported, all in subjects living in shared cells, and 4 of them required hospitalization. Table 1 displays the main characteristics of the study population. The mean age of the detained people was 42.9 years (range 19–76), the majority were males (90.5%) and Italians (87%), slightly more than half (54.6%) were married or cohabitant, 73.6% had sons or daughters, only 24.3% had obtained a high school or university degree, and 54.5% were employed before current detention. For fewer than half of participants (42.5%), this was the first episode of incarceration, 32% reported a working activity in the prison, and almost all (92.8%) lived in shared cells. Moreover, 34.8% of the detained people declared to be affected by chronic diseases, and, of these, 45.6% and 35.6% by cardiovascular and respiratory diseases, respectively, whereas previous SARS-CoV-2 infection was declared by 12.8% of the respondents.

### 3.2. Attitudes about COVID-19 Vaccination and Behaviors before and after COVID-19 Vaccination

High concern about contracting COVID-19 after vaccination was expressed by 47.1% of participants, whilst belief about usefulness and safety of COVID-19 vaccines was reported by 60.6% and 53.8% of respondents, with an average value of 7.5 and 7.2, respectively.

Adherence to the active offer of COVID-19 vaccination involved 89.7% of detained subjects. The majority of the sample received COVID-19 Janssen vaccine (Janssen Cilag) (57.9%), followed by Spikevax (Moderna) (25.5%), Comirnaty (Pfizer/BioNTech) (14.5%), and Vaxzevria (AstraZeneca) (2.1%). Moreover, 64.5% reported side effects after COVID-19 vaccination, and the most frequent were pain at the injection site (57.9%), fever (42.8%), asthenia (41.8%), bone, joint and muscular pain (33.6%), headache (26.4%), diarrhea (13%), dizziness (6.5%), nausea (6.2%), and vomiting (1.4%). No substantial differences in the frequency of side effects were reported between those with or without chronic diseases (65.2% vs. 64.1%). The most frequently reported reasons for vaccination uptake were being favorable to vaccinations (60.7%), perceiving oneself at risk of contracting COVID-19 (35.4%), trust in the effectiveness (28%) and the safety (22.2%) of the COVID-19 vaccine, trust in national authorities (16.8%), and in recommendations received by physicians (10.5%). Among those who refused COVID-19 vaccination, declared reasons were need of additional information (55.2%), lack of trust in national authorities (27.6%), concern about the safety (31%) and the effectiveness (13.8%) of COVID-19 vaccine, not being favorable to vaccinations (24.1%), and not perceiving oneself to be at risk of contracting COVID-19 (6.9%). Adherence to the influenza vaccination campaign in the 2020–2021 season was declared by 46.7% of responders.

Among COVID-19 vaccinated subjects, 32.5% reported to continue to always use face masks whenever needed, and 38.6% to always wash/disinfect their hands when required.

### 3.3. Univariate and Multivariate Regression Analysis

Among socio-demographic characteristics, only nationality (*p* = 0.027) was significantly associated with the COVID-19 vaccination uptake, whereas influenza vaccination uptake in the 2020–2021 season (*p* < 0.001), as well as belief that COVID-19 vaccine is safe (*p* = 0.018), physicians (*p* = 0.045) or media and newspapers (*p* = 0.030) as sources of information about COVID-19 vaccine, and need of additional information about COVID-19 vaccine (*p* = 0.023) were significant predictors of COVID-19 vaccination uptake in the univariate analysis (Table 2). Conversely, family and friends as sources of information about COVID-19 vaccine predicted a significant lower adherence to COVID-19 vaccine uptake (*p* = 0.048).

In multivariate analysis, COVID-19 vaccination uptake was significantly higher in females (OR = 15.94; 95% CI = 1.67–152.7), and in those who reported influenza vaccination uptake in the 2020–2021 season (OR = 6.21; 95% CI = 1.88–20.52), who had received information about COVID-19 vaccination from media and newspapers (OR = 4.37; 95% CI = 1.6–11.9), who did not express need of additional information about COVID-19 vaccine (OR = 0.29; 95% CI = 0.1–0.81), who believed that COVID-19 vaccine is safe (OR = 1.23; 95% CI = 1.03–1.47), and were involved in working activities in the prison (OR = 3.1; 95% CI = 1.03–9.36). Furthermore, education level showed to have an impact on COVID-19 uptake, since respondents with a primary school degree were significantly less likely to have undergone COVID-19 vaccination compared to those with a high school or university degree (OR = 0.31; 95% CI = 0.1–0.93) (Table 3).

### 3.4. Sources of Information about COVID-19 Vaccine

The most frequently mentioned sources of information about COVID-19 were media and newspapers (77.1%), followed by physicians for 51.6% of participants and family or friends (21.6%). Moreover, 47.3% of respondents reported they would benefit from additional information about COVID-19 vaccine.

## 4. Discussion

There is no doubt that the extent of acceptance of vaccines in the target populations is one of the most powerful indicators of the success or failure of vaccination strategies. This study has investigated the COVID-19 vaccination uptake in detained subjects, demonstrating a very high acceptance, with adherence to vaccination almost reaching 90% of the eligible population. Data reported in the literature account for very variable frequencies of uptake in the different countries; specifically, the WHO for the European Region has recently published a report of data obtained from voluntary submissions of Member States to the WHO Minimum Dataset Reporting System for places of detention, containing also cumulative vaccination coverage, with lowest adherence reported in Moldova (36.7%), as of August 2021, and Greece (57.6%), and highest in Spain (85.7%) and Poland (90%), as of October 2021 [1]. Outside Europe, Hagan et al. have reported 64.2% uptake in a study of the US Bureau of Prison up to April 2021 [31], Chin et al. 64.2% in California [25], and Berk et al. 77.7% in Rhode Island, up to February 2021 [32]. It is noteworthy, however, that these studies refer to different phases of the vaccination campaigns, and that different strategies have been adopted for the detained subjects in the involved countries; therefore, comparisons should take into consideration that different coverage achievements might have derived from all these differences. One remarkable result is the very large adherence in detained subjects also in comparison to the Italian general population, which, in the considered time period, reached a cumulative 82.5% coverage [33], thus, showing a higher level of acceptance of COVID-19 vaccination in this disadvantaged group. This finding is even more surprising when compared to detained subjects’ willingness to undergo a future COVID-19 vaccination, declared by 64% of them, in a survey conducted in the same area from March to April 2021 [21]. In the same study, a lack of self-confidence in the ability to protect themselves from SARS-CoV-2 infection in the prison setting was declared by a large majority of the participants and this perception may have prompted the decision to undergo COVID-19 vaccination [21]. Contribution to this high vaccination coverage may have also been related, as reported by Hagan et al. [31], to the relative absence of barriers for the accomplishment of vaccinations, which were offered by healthcare providers, and were performed on-site, without all the potential obstacles encountered by the general population for scheduling appointments, queueing in vaccination hubs, etc., which may have caused delays or refusal of vaccination.

It is worth underlying that the most frequently reported reasons for refusing COVID-19 vaccination were need of more information, followed by safety concerns about the vaccine and distrust in institutions; these findings may be very helpful for the development of interventions aimed at reducing barriers to COVID-19 vaccination, since they suggest that to build trust and vaccine confidence, educational programs on the effectiveness and safety of COVID-19 vaccines should involve trustworthy sources of information, such as healthcare workers [34,35,36,37] or peer educators, establishing programs which have been successful in promoting prevention activities, where detained people were trained as educators [38,39].

Most of the results on adherence to COVID-19 vaccination in incarcerated subjects were obtained from surveillance systems or registries; therefore, in most cases only distribution according to age, sex, and few other characteristics are reported to describe differences between those who accepted and those who refused to be vaccinated. Instead, this survey explored, in detail, several characteristics of incarcerated subjects that could influence the adherence to COVID-19 vaccination, and, indeed, significant differences were found in the two groups, which have implications for designing interventions to enhance COVID-19 vaccination in detained subjects. The main findings on predictors of COVID-19 vaccination uptake underline the role of gender and education with females and more educated subjects being more likely to be vaccinated, as well as those who believe that COVID-19 vaccine is safe, have undergone influenza vaccination, and have a working activity in prison. Moreover, a significantly higher coverage was found in detained subjects who reported to have received information from media and newspapers, whereas, as expected, those who had been vaccinated declared to have no more need of information on COVID-19 vaccination. Similar studies have revealed that socio-demographic characteristics are associated with COVID-19 uptake, but coverage has been found to be higher for detained males [31], whereas higher education has been found to be a predictor of higher adherence to vaccinations in several settings and populations [40,41]. Moreover, safety issues on vaccines are one of the most relevant arguments for vaccine hesitancy [26,42], remarking the need to improve communication on benefits and risks of COVID-19 vaccination in the hesitant subjects, whereas a favorable attitude towards vaccinations is generally related to acceptance of various vaccinations, as demonstrated, in this case, by the association between influenza and COVID-19 vaccination uptake. The finding that working activity in prison was a predictor of COVID-19 vaccination uptake is not surprising, since in previous studies, detained subjects involved in working activities rated themselves to be in good health status [24] and reported high self-confidence in their ability to protect themselves from COVID-19 infection [21], suggesting that working activity is associated to a positive attitude towards healthy behaviors, such as, in this case, COVID-19 vaccination uptake. The findings on the predictors of adherence to COVID-19 vaccination allow to design the profile of the detained subjects that are more likely to refuse COVID-19 vaccine and that should be the target of interventions aimed at promoting adherence. They are more likely to be low educated males, who do not work inside prison, who are not very confident on the safety of COVID-19 and other vaccines, who do use media to get information about vaccination, but that would like to be more informed.

There is consensus that detention may be viewed as an opportunity for health needs assessment and provision of prevention, diagnostic, and treatment activities in detained people [43,44], but it has been reported that guidelines and policies on prison health do not adequately reflect the relevance of prevention of communicable diseases through vaccination [45]; this is confirmed by published literature indicating that incarcerated people are under-immunized, particularly against HBV, influenza, MMR, and pneumococcus [45,46]. In this context, the results of this study are of relevance, since they showed the feasibility and success of interventions promoting vaccination in the achievement of very high coverage in this underserved population. These findings are in line with those of a recent systematic review investigating vaccinations in prison, which highlighted a very sparse literature on prison vaccination interventions, but revealed that, especially for HBV vaccinations, very high coverage rates were reached when interventions were promoted in detained subjects [45].

To appreciate the results of this survey, some potential limitations in the design and the implementation of the research need to be addressed. First, this study adopted a cross sectional research design and, thus, it did not allow analysis of the direction of influence between the different variables and COVID-19 vaccine uptake. Second, although appropriate methods for sampling have been employed, the current study remains limited by the use of data collected only from those who decided to enroll in the study. However, since the large majority of detained subjects accepted to participate with a very high response rate, it is plausible to argue that the results of the survey would have not been substantially changed by the inclusion of those who refused to participate. Third, there was potential bias attributable to the use of a self-reporting instrument, and we were unable to accurately determine vaccine uptake. Fourth, the participating detained people were recruited from three prisons in southern Italy, which mostly hosted males; therefore, the study has examined only a subset of detained people, potentially limiting the generalizability of the results to the wider population of detained people in Italy. Finally, participants’ awareness of being part of a research study may have influenced behaviors, potentially overestimating the socially desirable behaviors. Based on these limitations, future research should include women and institutions located in a larger geographic area and should assess adherence to vaccination through more objective tools, such as vaccination records. Moreover, follow up of vaccinated and unvaccinated subjects could add knowledge on the real-world effectiveness of COVID-19 vaccines in an overcrowded and challenging context, as well as on the impact of the interventions implemented to counter barriers and to promote vaccinations.

## 5. Conclusions

This study revealed a high self-reported COVID-19 vaccination coverage in detained subjects, supporting the effectiveness of the strategy aimed at giving priority to COVID-19 vaccinations in prisons. Success of the campaign appears to be related to the on-site availability of vaccines and on the involvement of HCWs. Further efforts are needed to contrast the hesitancy of those who refused vaccination by promoting clear communication on risks and benefits of vaccinations and by involving HCW and peer educators.

## Figures and Tables

**Table 1 vaccines-10-00673-t001:** Characteristics of the study population (*n* = 517).

Characteristics		
Socio-Demographics	*n*	%
Age, years ª	42.9 ± 11.47 (19–76) *	
19–37	169	32.9
38–47	171	33.3
>47	174	33.8
Gender		
Male	468	90.5
Female	49	9.5
Marital status ª		
Married/cohabitant	270	54.6
Unmarried/widowed/separated/divorced	225	45.4
Nationality ª		
Italians	449	87
Foreigners	67	13
Sons/daughters ª		
No	135	26.4
Yes	377	73.6
Education level ª		
Primary school	106	22
Middle school	258	53.7
High school or university degree	117	24.3
Occupation before detention ª		
Unemployed	230	45.5
Employed	275	54.5
**Detention**		
Institution		
Prison 1	292	56.5
Prison 2	129	24.9
Prison 3	96	18.6
First detention ª		
No	270	57.5
Yes	200	42.5
Length of detention, months ª	91.6 ± 83.49 (2–462) *	
<60	138	46.1
≥60	161	53.9
Working activity in the prison ª		
No	347	68
Yes	163	32
Type of cell ª		
Individual	28	7.2
Shared	362	92.8
**Anamnestic**		
At least one chronic disease		
No	337	65.2
Yes	180	34.8
Cardiovascular diseases	82	45.6
Respiratory diseases	64	35.6
Metabolic diseases	40	22.2
Genitourinary diseases	29	16.1
Autoimmune diseases	17	9.4
Neurological diseases	15	8.3
Obesity (BMI > 35)	12	6.7
Oncological diseases	10	5.6
Other	7	3.9
Hematological diseases	2	1.1
Influenza vaccination uptake in the 2020–2021 influenza season ª		
No	310	60.2
Yes	205	39.8
Pneumococcal vaccination uptake ª		
No	485	94.7
Yes	27	5.3
Previous SARS-CoV-2 infection		
No	451	87.2
Yes	66	12.8
COVID-19 vaccine uptake		
No	53	10.3
Yes	464	89.7
Side effects after COVID-19 vaccine ª^,^º		
No	162	35.5
Yes	294	64.5
**Attitudes towards COVID-19 vaccination**		
Concern about contracting COVID-19 after vaccination ª^,^º	6.46 ± 3.02 (1–10) *	
Low (1–7)	241	52.9
High (8–10)	215	47.1
Belief that COVID-19 vaccine is useful ª	7.5 ± 2.37 (1–10) *	
Low (1–7)	194	39.4
High (8–10)	298	60.6
Belief that COVID-19 vaccine is safe ª	7.2 ± 2.34 (1–10) *	
Low (1–7)	228	46.2
High (8–10)	265	53.8
**Behaviors after COVID-19 vaccination º**		
Wearing mask ª		
Never	19	4.1
Rarely	42	9.1
Sometimes	62	13.5
Often	188	40.8
Always	150	32.5
Hands antisepsis ª		
Never	5	1.1
Rarely	20	4.4
Sometimes	48	10.5
Often	208	45.4
Always	177	38.6
**Sources of information about COVID-19 vaccine**		
Physicians ª		
No	243	48.4
Yes	259	51.6
Media and newspapers ª		
No	115	22.9
Yes	387	77.1
Family and friends ª		
No	392	78.1
Yes	110	21.9
Need of additional information about COVID-19 vaccine ª		
No	261	52.7
Yes	234	47.3

* Mean ± Standard deviation (range). ª Number for each item may not add up to total number of study population due to missing values. º Among vaccinated subjects.

**Table 2 vaccines-10-00673-t002:** COVID-19 vaccination uptake according to several population characteristics (*n* = 517).

Characteristics	Total	COVID-19 Vaccine Uptake *n* = 464 (89.7%)	
Socio-Demographics	*n*	*n*	%
Age, years ª			
19–37	169	154	91.1
38–47	171	153	89.5
>47	174	155	89.1
		χ^2^ = 0.441, 2 df, *p* = 0.802	
Gender			
Male	468	419	89.5
Female	49	45	91.8
		χ^2^ = 0.256, 1 df, *p* = 0.613	
Marital status ª			
Married/cohabitant	270	237	87.8
Unmarried/widowed/separated/divorced	225	207	92
		χ^2^ = 2.367, 1 df, *p* = 0.124	
Nationality ª			
Italians	449	408	90.9
Foreigners	67	55	82.1
		χ^2^ = 4.875, 1 df, *p* = 0.027	
Sons/daughters ª			
No	135	121	89.6
Yes	377	338	89.7
		χ^2^ ≤ 0.001, 1 df, *p* = 0.993	
Education level ª			
Primary school	106	92	86.8
Middle school	258	232	89.9
High School or university degree	117	106	90.6
		χ^2^ = 1.012, 2 df, *p* = 0.603	
Occupation before detention ª			
Unemployed	230	205	89.1
Employed	275	248	90.2
		χ^2^ = 0.149, 1 df, *p* = 0.699	
**Detention**			
Institution			
Prison 1	292	245	83.9
Prison 2	129	128	99.2
Prison 3	96	91	94.8
		χ^2^ = 26.085, 2 df, *p* ≤ 0.001	
First detention ª			
No	270	247	91.5
Yes	200	175	87.5
		χ^2^ = 1.986, 1 df, *p* = 0.159	
Length of detention, months ª			
	91.6 ± 83.49 (2–462) *	*t* test= 0.079, 447 df, *p* = 0.943	
<60		138	46.1
≥60		161	53.9
Working activity in the prison ª			
No	347	305	87.9
Yes	163	152	93.3
		χ^2^ = 1.898, 1 df, *p* = 0.065	
Type of cell ª			
Individual	28	28	100
Shared	362	328	90.6
		χ^2^ = 2.8810, 1 df, *p* = 0.090	
**Anamnestic**			
At least one chronic disease			
No	337	306	90.8
Yes	180	158	87.8
		χ^2^ = 1.166, 1 df, *p* = 0.280	
Influenza vaccination uptake in the 2020–2021 influenza season ª			
No	310	265	85.5
Yes	205	197	96.1
		χ^2^ = 15.057, 1 df, *p* ≤ 0.001	
Pneumococcal vaccination uptake ª			
No	485	434	89.5
Yes	27	25	92.6
		χ^2^ = 0.266, 1 df, *p* = 0.606	
Previous SARS-CoV-2 infection			
No	451	406	90
Yes	66	58	87.9
		χ^2^ = 0.287, 1 df, *p* = 0.592	
**Attitudes towards COVID-19 vaccine**			
Belief that COVID-19 vaccine is useful ª			
1–10	7.5 ± 2.37 (1–10) *	*t* test= −1.379, 490 df, *p* = 0.168	
Belief that COVID-19 vaccine is safe ª			
1–10	7.2 ± 2.34 (1–10) *	*t* test= −2.371, 491 df, *p* = 0.018	
**Sources of information about COVID-19 vaccine**			
Physicians ª			
No	243	217	89.3
Yes	259	244	94.2
		χ^2^ = 4.027, 1 df, *p* = 0.045	
Media and newspapers ª			
No	115	100	87
Yes	387	361	93.3
		χ^2^ = 4.729, 1 df, *p* = 0.030	
Family and friends ª			
No	392	365	93.1
Yes	110	96	87.3
		χ^2^ = 3.905, 1 df, *p* = 0.048	
Need of additional information about COVID-19 vaccine ª			
No	261	245	93.9
Yes	234	206	88
		χ^2^ = 5.188, 1 df, *p* = 0.023	

* Mean ± Standard deviation (range). ª Number for each item may not add up to total number of study population due to missing values.

**Table 3 vaccines-10-00673-t003:** Multivariate logistic regression analysis to characterize factors associated with COVID-19 vaccine uptake.

	OR *	SE **	95% CI º	*p*
Model 1. COVID-19 Vaccine Uptake				
Log likelihood = −69.056455; χ^2^ = 68.41 (12 df); *p* ≤ 0.001				
Influenza vaccination uptake in the 2020–2021 influenza season				
No	1 ª			
Yes	6.21	3.79	1.88–20.52	0.003
Having received information about COVID-19 vaccination from media and newspapers				
No	1 ª			
Yes	4.37	2.23	1.6–11.9	0.004
Gender				
Male	1 ª			
Female	15.94	18.38	1.67–152.7	0.016
Need of additional information about COVID-19 vaccine				
No	1 ª			
Yes	0.29	0.15	0.1–0.81	0.019
Belief that COVID-19 vaccine is safe				
No	1 ª			
Yes	1.23	0.11	1.03–1.47	0.023
Education level				
High school or university degree	1 ª			
Primary school	0.31	0.17	0.1–0.93	0.036
Working activity in the prison				
No	1 ª			
Yes	3.1	1.75	1.03–9.36	0.044
Having sons/daughters				
No	1 ª			
Yes	2.28	1.13	0.86–6.03	0.096
Age, years				
>47	1 ª			
38–47	0.59	0.29	0.22–1.54	0.280
First detention				
No	1 ª			
Yes	1.63	0.78	0.63–4.19	0.312

* Adjustment for institution (prison) was performed; ** Standard error; **º** Confidence interval; ª Reference category.

## Data Availability

The data that support the findings of this study are available from the corresponding author, [M.P.], upon reasonable request.

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
