# Peer review of "COVID-19 Vaccination Uptake and Related Determinants in Detained Subjects in Italy"

_vaccines, 2022, doi:10.3390/vaccines10050673_

Round 1

Reviewer 1 Report

The manuscript by Gabriella Di Giuseppe and colleagues titled “COVID-19 vaccination uptake and related determinants in detained subjects in Italy” described a unique study on the detained population about the COVID-19 vaccination uptake. While the manuscript represents an interesting finding that 90% of the eligible population has very high acceptance and adherence to vaccination. However, there are major considerations that prevent the manuscript’s publication in its current form.

Major considerations:

  1. The author has not given the proper statistical importance to this work. Please supplement the introduction with the proper statistics like the percentage of COVID outbreaks in the prison, % of mortality, % of hospitalization/recovery, etc across the world and/ in Italy.
  2. Line 45-49: How do you defend this argument that the people in the places of detention are highly vulnerable to COVID? It would be more convenient for the readers if you could supplement your argument with proper statistics/details like what is the population density of prisoners per prison, the average number of prisoners entry/exiting in the given time, etc
  3. What is the important source or backbone for the designing of your questionnaire?
  4. Line 153: “whereas previous SARS-CoV-2 infection was declared by 12.8% of the respondents” Are there any data showing the SARS-CoV-2 infection happened in the prison during detention since the majority of them were lived in a shared cell?
  5. The majority of the prisoners are having middle school level education; how do you justify the survey which can represent the majority of the population with a higher level of education?
  6. The author has not described, what are the side effects seen in the vaccinated individuals. Are these side effects were due to vaccinations or due to preexisting health conditions? These questions will play a major role in finding out the possible reasons for the rejection of vaccination.
  7. The author has not described the type of SARS-CoV-2 vaccines, the manufacturer of the vaccine, the clinically proven efficacy of the vaccine during the clinical trials, etc. This information will be very useful to analyze the outcome of the survey effectively.
  8. The author can also elaborate on the major factors involved in the vaccine uptake based on their p values from Table 3.
  9. Line 177-183: proper alignment of the text is needed.
  10. The future direction based on your survey can be mentioned to overcome the shortcomings of your survey. 

Author Response

The author has not given the proper statistical importance to this work. Please supplement the introduction with the proper statistics like the percentage of COVID outbreaks in the prison, % of mortality, % of hospitalization/recovery, etc across the world and/ in Italy.

As suggested, Statistics on COVID-19 frequency, mortality and hospitalization in prisons in Italy and in Europe have been included.

Line 45-49: How do you defend this argument that the people in the places of detention are highly vulnerable to COVID? It would be more convenient for the readers if you could supplement your argument with proper statistics/details like what is the population density of prisoners per prison, the average number of prisoners entry/exiting in the given time, etc

As suggested, characteristics of prison overcrowding and turnover have been added

What is the important source or backbone for the designing of your questionnaire?

As suggested, sources of the questionnaire have been provided.

Line 153: “whereas previous SARS-CoV-2 infection was declared by 12.8% of the respondents” Are there any data showing the SARS-CoV-2 infection happened in the prison during detention since the majority of them were lived in a shared cell?

As, suggested, data on occurrence of COVID-19 cases in the selected prisons have been provided in the results section.

The majority of the prisoners are having middle school level education; how do you justify the survey which can represent the majority of the population with a higher level of education?

In response to this point, this result is not surprising since in Italy education is mandatory up to 16 years of age and indeed middle schools end at age 14-15; therefore this result is expected also in disadvantaged population groups.

The author has not described, what are the side effects seen in the vaccinated individuals. Are these side effects were due to vaccinations or due to preexisting health conditions? These questions will play a major role in finding out the possible reasons for the rejection of vaccination.

In response to this point, this was a survey and information was provided by incarcerated subjects. We have now included information on reported side-effects of vaccination, and have tried to verify whether these effects differed according to underlying medical conditions.

The author has not described the type of SARS-CoV-2 vaccines, the manufacturer of the vaccine, the clinically proven efficacy of the vaccine during the clinical trials, etc. This information will be very useful to analyze the outcome of the survey effectively.

As suggested, more information on the characteristics of the type of SARS-CoV-2 vaccines used have been included.

The author can also elaborate on the major factors involved in the vaccine uptake based on their p values from Table 3.

In response to this point, a general discussion on significantly associated factors (p<0.05) to adherence to COVID-19 vaccination has been reported (Lines…). We have now elaborated, based on these factors, the profile of incarcerated subjects who are more likely to refuse COVId-19 vaccination.

Line 177-183: proper alignment of the text is needed.

As suggested, alignment of the text has been done

The future direction based on your survey can be mentioned to overcome the shortcomings of your survey.

As suggested, we have more thoroughly discussed on direction of future research based on survey limitations.

Reviewer 2 Report

In the manuscript presented here, the authors describe the results of a socio-medical study of SARS-CoV-2 vaccination in prison inmates. Vaccination acceptability, as well as positive and negative influencing factors, are considered. 
Studies on vaccination acceptance have gained particular importance worldwide in the context of the COVID 19 pandemic. Findings, especially on the willingness to be vaccinated among socially marginalized groups, are of great importance. However, they are not always in the focus of public or scientific attention. In this respect, the study presented here is of particular relevance.
The presentation of the study methodology and the results obtained are comprehensive and coherent. They are suitable to enable the interpretation of the data also in comparison with other similar studies. I have no substantial objections to the planning and analysis. Also the presentations in the Discussion section of the manuscript are well-balanced and based on the collected data. However, the manuscript should be reviewed again by a native English speaker before final publication.
Otherwise, I have no significant comments or suggestions for changes. From my point of view, there should be no obstacles to a timely publication in Vaccines.

Author Response

The manuscript should be reviewed again by a native English speaker before final publication.

As suggested, the manuscript has undergone review by a native English speaker.

Round 2

Reviewer 1 Report

The author has answered all the comments. I am satisfied with the revised version of the manuscript. This can be published in its original form after checking for minor English grammar and spelling.